Knowledge, attitude, and practice toward perioperative neurocognitive disorders among healthcare workers in Shandong, China: a cross-sectional study

Zhang Ke 1
Gu Xuefeng 2
Li Xiaoru 2
Peng Honghai penghh5463@163.com 2
1 Department of Anesthesia, Central Hospital Affiliated to Shandong First Medical University , Jinan , Shandong , China
2 Department of Neurosurgery, Central Hospital Affiliated to Shandong First Medical University , Jinan , Shandong , China
Parker Matthew
Electronic publication date: 2025 Dec 9
Publication date: 2025
Volume: 13
Electronic Location ID: e20450
Received 2025 May 19; Accepted 2025 Oct 31
Copyright: ©2025 Zhang et al.
Copyright year: 2025
Copyright holder: Zhang et al.
License: This is an open access article distributed under the terms of the Creative Commons Attribution License, which permits unrestricted use, distribution, reproduction and adaptation in any medium and for any purpose provided that it is properly attributed. For attribution, the original author(s), title, publication source (PeerJ) and either DOI or URL of the article must be cited.
License URL: https://creativecommons.org/licenses/by/4.0/

Keywords: Knowledge, Attitude, Practice, Perioperative neurocognitive disorders, China, Cross-sectional study

Funding: The authors received no funding for this work.

==============================
Background

Perioperative neurocognitive disorders (PND) refer to neurocognitive abnormalities detected during the perioperative period, currently including postoperative delirium, delayed neurocognitive recovery, postoperative neurocognitive disorder, and both mild and major cognitive impairments. It is a prevalent complication among surgical patients, particularly in older populations. Moreover, PND can have a profound effect on the quality of life for these patients. PND has been previously reported to occur with a prevalence of 10–54% during the first few weeks following surgery, and when persistent at the time of discharge or in the longer term, PND may increase mortality. Therefore, it is of considerable importance for healthcare professionals to accurately identify the occurrence of PND and implement appropriate management measures. This research aims to investigate the knowledge, attitude, and practice (KAP) of healthcare workers concerning perioperative neurocognitive disorders (PND) in Shandong Province, China.

Methods

This cross-sectional study recruited healthcare workers from twelve hospitals in Shandong Province, China, utilizing self-administered questionnaires distributed between February and May 2025. There are 12 items in the knowledge dimension, and the total score range for this dimension is 0 to 12 points. There are a total of seven items in the attitude dimension, and the total score range for this dimension is 7 to 35 points. There are a total of nine items in the practice dimension, and the total score range for this dimension is from 9 to 45 points.

Results

The analysis included a total of 309 valid questionnaires, which represents an impressive 87.78% response rate. Among them, a total of 193 doctors (62.5%) and 116 nurses (37.5%) participated in the survey, respectively. The PND knowledge score was 9.14 (SD = 2.85, range = 0–12). The PND attitudes score was 30.89 (SD = 3.81, range = 7–35). The PND practices score was 36.16 (SD = 7.77, range = 9–45). Furthermore, Path analysis indicated that participants’ knowledge of PND among healthcare workers had a direct and positive effect on their attitudes (β = 0.545, P < 0.001) and practices towards PND (β = 0.230, P = 0.002). And attitudes towards PND also had a direct and positive effect on practices (β = 0.238, P = 0.001). In addition, the knowledge exerted an indirect effect on practice through attitudes (β = 0.581, P = 0.003).

Conclusions

Healthcare practitioners have sufficient knowledge, and they demonstrate positive attitudes and adopt proactive approaches regarding PND. The understanding and perspectives of healthcare workers are closely linked to their actions towards PND. Educational training is pivotal in shaping their knowledge, attitude, and practice.

Introduction

Perioperative neurocognitive disorders (PND) refer to neurocognitive abnormalities detected during the perioperative period, currently including postoperative delirium, delayed neurocognitive recovery, postoperative neurocognitive disorder, and both mild and major cognitive impairments (Dilmen et al., 2024; Kong, Xu & Wang, 2022). Delirium is a syndrome characterized by a sudden and fluctuating change in attention, consciousness level, and cognitive function. It can occur before surgery, but is most commonly seen within 7 days after the operation. Delayed neurocognitive recovery indicates new cognitive decline within 30 days after the surgery, and many or even most patients fully recover from the early postoperative cognitive impairment. Postoperative cognitive impairment refers to the decline in cognitive function detected from 30 days after the surgery to 12 months of follow-up. Characterized by impaired neuropsychological function, PND manifests disturbances in memory, consciousness, and attention, along with alterations in personality, and is one of the most common postoperative complications in older adult patients (Shen et al., 2020; Xu et al., 2021). A previous study reported that a 65-year-old male, who presented four weeks after undergoing total knee arthroplasty, was diagnosed with perioperative neurocognitive disorders (PND). The condition was manifested by acute confusion and agitation lasting for 19 days. Once PND occurs, it will compromise patient autonomy, reduce quality of life, extend hospital stays, and increase morbidity and mortality rates, especially impacting older adult patients (Dilmen et al., 2024). PND has been previously reported to occur with a prevalence of 10–54% during the first few weeks following surgery, and 12–17% at 3 months, with the number decreasing to 3% by 12 months. When persistent at the time of discharge or in the longer term, PND may increase mortality (Yang et al., 2022).

Previous studies have shown that avoiding the use of benzodiazepines and anticholinergic drugs before surgery and receiving cognitive training can reduce the incidence of perioperative neurocognitive disorders (PND). Reasonable selection of anesthetic drugs and anesthesia types during the operation, as well as monitoring of cerebral oxygen saturation, can also help reduce the occurrence of PND. In addition, promoting sleep and implementing enhanced recovery after surgery (ERAS) strategies after the operation further reduced the risk of PND (Duprey et al., 2022; Hughes et al., 2020; Li et al., 2021; O’Gara et al., 2020; Wang et al., 2021; Yang et al., 2023). Among them, enhanced recovery after surgery (ERAS) refers to a series of evidence-based perioperative nursing interventions designed to attenuate the stress response to surgery in the perioperative period and hasten recovery (Ljungqvist et al., 2021). The prevention and treatment of PND is a clinical issue that runs throughout the entire perioperative period and requires multidisciplinary collaboration, including nurses, surgeons, anesthesiologists, and personnel in the operating room, etc. Medical workers of different roles play unique and irreplaceable roles at different stages of patient treatment. Among them, anesthesiologists are responsible for formulating and implementing the anesthesia plan during the operation (such as anesthesia methods and drug selection, anesthesia depth monitoring, avoiding the use of drugs that cause delirium, blood pressure, and body temperature management), which is an important part of preventing PND (Hu et al., 2024; Luo et al., 2024). Surgeons participate in the selection of surgical methods, the control of operation duration, and the decision-making on intraoperative bleeding and blood transfusion, all of which are directly related to the size of surgical stress, and stress is an important trigger of PND. Nurses are responsible for preoperative cognitive assessment and screening, postoperative pain management, sleep improvement, and early mobilization, which are all key aspects of non-pharmacological prevention (Zhou et al., 2022). Considering the worldwide accelerated aging of the population, an increasing number of operations, and consequent increasing incidence of PND, healthcare workers must be aware of PND and take joint measures to prevent PND during the perioperative period.

The knowledge-attitude-practice (KAP) model serves as a significant framework within the healthcare sector, particularly for understanding the intricate relationship between individual knowledge and beliefs and their subsequent effects on health behaviors. This model is instrumental in numerous studies related to disease management, emphasizing that the level of knowledge possessed by healthcare professionals or patients plays a crucial role in shaping their attitudes toward diagnosing and treating diseases (Cai et al., 2024; Peng et al., 2024; Zhou et al., 2022).

The KAP theory posits that changes in human behavior can be achieved through a sequential process involving the acquisition of knowledge, the formation of beliefs, and the establishment of behaviors (Kharaba et al., 2021). Consequently, these attitudes profoundly impact their actual behavioral performance in clinical settings, ultimately affecting the quality of medical care provided and the health outcomes achieved. By examining this model, researchers can gain valuable insights into how educational interventions might improve both knowledge and attitudes, thereby fostering better health practices among individuals.

However, most previous studies have primarily focused on the role of anesthesiologists in the occurrence of PND while neglecting the involvement of other surgery-related healthcare workers (Hu et al., 2024; Luo et al., 2024). Therefore, this study aimed to explore the KAP toward PND for healthcare workers in Shandong Province, China.

Materials & Methods

Study design and participants

Healthcare professionals from Shandong Province, China, participated in this cross-sectional study conducted between February and May 2025. The selection criteria for participants included several specific requirements: first, candidates had to be medical staff working in departments that directly interact with surgical patients. Second, they needed to possess a valid Chinese medical or nursing qualification certificate and have a minimum of six months of clinical experience. Lastly, Participants were informed that participation was entirely voluntary, and they could withdraw from the study at any time without any penalty. Consent was implied upon completion and submission of the questionnaire. Individuals who lacked the necessary qualifications, were undergoing training or education, or who did not consent to participate were deliberately excluded from the study. Our study was conducted and reported in accordance with the STROBE guidelines for observational studies. The research received ethical approval from the Ethics Committee of Central Hospital Affiliated to Shandong First Medical University, bearing the approval number 202501200008. Furthermore, all participants provided written informed consent before their involvement in the study.

Clinical guidelines and previous studies were consulted when designing the self-administered questionnaire (Dilmen et al., 2024; Kong, Xu & Wang, 2022; Luo et al., 2024). A small-scale pilot test was conducted involving 39 healthcare workers to evaluate the internal consistency of the questionnaire using Cronbach’s α. The obtained Cronbach’s α value of 0.879 indicates a high level of reliability for the instrument employed in this study. The finalized questionnaire, presented in Chinese, comprised a total of 40 items organized into four distinct dimensions. Specifically, twelve items were designed to capture the basic characteristics of the subjects, while the dimensions related to KAP included 12, seven, and nine items, respectively, ensuring a comprehensive assessment across these critical factors.

The knowledge dimension is structured around a total of 12 questions, where participants earn 1 point for each correct response, while incorrect or uncertain answers yield a score of 0 points. Consequently, the scoring for this dimension ranges from a minimum of 0 points to a maximum of 12 points. This scoring system facilitates a straightforward assessment of participants’ knowledge levels, providing a clear numeric representation of their understanding of the knowledge of PND. In contrast, the attitude dimension comprises seven questions evaluated using a 5-point Likert scale. Questions that elicit positive attitudes are scored such that ‘strongly agree’ equates to five points and ‘strongly disagree’ equates to one point. Conversely, negative attitude questions are scored inversely, meaning that a response of ‘strongly agree’ would receive one point, while a response of ‘strongly disagree’ would score five points. Thus, the total possible score for this dimension ranges from 7 to 35 points, effectively capturing the breadth of respondents’ attitudes toward the PND. Lastly, the practice dimension consists of nine questions and employs a 5-point Likert scale. In this case, responses vary from ‘always’ (five points) to ‘never’ (one point). The scores for this dimension can thus range from a minimum of 9 points to a maximum of 45 points, providing a quantitative measure of the frequency and consistency of positive practices related to the PND. To establish benchmarks for evaluating the effectiveness of knowledge, attitude, and practice, a threshold has been set at a total score of 70% or higher for each dimension. Achieving this threshold indicates adequate knowledge, a constructive attitude, and engaged practice among participants, thereby highlighting their proficiency and readiness in the prevention and treatment of PND (Lee & Suryohusodo, 2022).

The electronic questionnaires were constructed using WPS Office software (Kingsoft Office Software, Beijing, China), and WeChat (Tencent, Shenzhen, China) was used to distribute the online questionnaire to the participants. As there were approximately 6,000 potential researchers in 12 hospitals, a convenience sampling was conducted. Utilizing convenient sampling, electronic questionnaires were disseminated to healthcare workers in the surgery-related department through WeChat groups by the leaders of the twelve hospitals. They were encouraged by their leaders to participate in the research, ensuring that the questionnaire was distributed exclusively to individuals who met the specified eligibility criteria.

To ensure the confidentiality of participants, anonymity was maintained throughout the survey. At the onset of the questionnaire, participants were required to indicate their consent by selecting ‘yes’ in response to the statement, ‘I acknowledge and agree that the collected data will be used for scientific research.’ Individuals who did not select the ‘yes’ option effectively opted out of consenting to participate in the study, thereby preventing them from completing the questionnaire. To uphold the integrity and quality of the collected data, stringent measures were implemented regarding the submission of responses. Each individual was permitted only one submission based on their unique IP address, which helped to avoid duplicate entries and maintain a high standard of data quality. Furthermore, participants were required to answer all items in the questionnaire. Considering the average response time for the small-scale pilot test is 90 s. Submissions completed in less than 90 s or exhibiting logical inconsistencies were excluded from the analysis. Additionally, responses where participants consistently selected the same answer across any section of the KAP questionnaire were deemed invalid and removed from consideration in the final results.

Sample size

The optimal sample size should be a minimum of tenfold the number of predictors (Cai et al., 2024). Given that this questionnaire contains 28 independent variables, the minimum required sample size would be 280. Accounting for a 20% non-response rate, the final required sample size was 336.

Statistical analysis

Data analysis utilized SPSS version 26.0 (IBM, Armonk, NY, USA). Continuous variables were represented as mean ± standard deviation (SD) and compared through analysis of variance (ANOVA). Categorical variables were presented as n (%). The Pearson correlation analysis was employed to examine the relationships among the KAP dimensions. The Structural Equation Modeling (SEM) was used to validate the impact of knowledge on attitudes and practices, as well as the influence of attitudes on practices, which was analyzed by AMOS 24.0 (IBM, NY, USA). A two-sided significance level of P < 0.05 was deemed statistically significant.

Results

Out of the 352 questionnaires collected for this study, 43 were deemed invalid and subsequently excluded from the analysis. This exclusion included 28 respondents who provided identical answers across all items in the KAP questionnaire, indicating a lack of genuine engagement with the survey. Additionally, ten respondents were removed due to their exceptionally brief response time of less than 90 s, which raised concerns regarding the accuracy and reliability of their answers. Finally, five cases were excluded as the individuals had not obtained a Chinese medical or nursing qualification certificate. Ultimately, this rigorous screening process resulted in the inclusion of 309 valid questionnaires, yielding an effective questionnaire response rate of 87.78%.

Table 1 presents a comprehensive overview of the demographic characteristics and KAP scores of the 309 participants involved in the study. Among these participants, a notable 60.5% were identified as female, indicating a slight female majority in the participants. The average age of participants was calculated to be 38.50 years with a standard deviation of 10.20 years, highlighting a diverse range of ages within the cohort, with 40.5% of participants being over 40 years old. This age distribution may contribute to the varied experiences and perspectives among the individuals surveyed. In terms of Occupation type, a significant portion of the participants, approximately 62.5%, were practicing physicians. Additionally, 33.7% of these professionals reported having 10 to 20 years of experience in their respective specialties, suggesting a considerable level of expertise within the sample. Despite this wealth of knowledge and experience, it is noteworthy that only 29.1% of the participants had engaged in training related to PND. These results indicate that the training of PND still lacks in terms of popularity and participation among healthcare workers. The mean KAP scores, which serve as indicators of the participants’ PND, were recorded as follows: 9.14 ± 2.85 (range: 0 to 12) for knowledge, 30.89 ± 3.81 (range: 7 to 35) for attitudes, and 36.16 ± 7.77 (range: 9 to 45) for practices. These scores indicate a varied knowledge, attitude, and practice of concepts related to PND among the participants, suggesting that enhancing both knowledge, attitude, and practice in the prevention and treatment of PND is of paramount importance.

Table 1 Demographic characteristics and KAP scores of the participants.

Variables	N (%)	Knowledge score
Mean ± SD	P	Attitude score
Mean ± SD	P	Practice score
Mean ± SD	P	
Total	309	9.14 ± 2.85		30.89 ± 3.81		36.16 ± 7.77		
Gender			0.289		0.220		0.403	
Male	122(39.5)	8.93 ± 3.09		30.56 ± 4.02		35.70 ± 8.17		
Female	187(60.5)	9.28 ± 2.68		31.10 ± 3.66		36.45 ± 7.51		
Age (years)			0.017		0.002		0.275	
<30	68(22.0)	9.07 ± 2.84		29.51 ± 4.33		37.43 ± 7.06		
[30, 40]	116(37.5)	9.70 ± 2.53		31.53 ± 3.58		36.06 ± 8.08		
>40	125(40.5)	8.66 ± 3.05		31.04 ± 3.54		35.55 ± 7.82		
Residency			0.061		0.172		0.500	
Urban	233(75.4)	8.97 ± 2.94		31.06 ± 3.74		36.33 ± 7.45		
Non-urban	76(24.6)	9.67 ± 2.47		30.37 ± 3.99		35.63 ± 8.70		
Education			0.130		0.006		0.289	
Associated Degree and lower	22(7.1)	8.00 ± 3.72		28.86 ± 4.49		35.00 ± 9.34		
Bachelor’s Degree	210(68.0)	9.17 ± 2.65		30.80 ± 3.75		36.63 ± 7.68		
Master’s Degree and higher	77(24.9)	9.38 ± 3.06		31.71 ± 3.56		35.18 ± 7.51		
Occupation type			0.075		<0.001		0.334	
Physician	193(62.5)	9.36 ± 2.79		31.67 ± 3.45		35.82 ± 7.46		
Nurse	116(37.5)	8.77 ± 2.91		29.58 ± 4.02		36.71 ± 8.27		
Professional title			0.738		0.007		0.792	
Junior and below	97(31.4)	8.99 ± 2.99		30.01 ± 4.29		36.57 ± 8.34		
Intermediate	112(36.2)	9.29 ± 2.94		30.91 ± 3.71		35.83 ± 7.47		
Vice senior and above	100(32.4)	9.11 ± 2.61		31.71 ± 3.22		36.12 ± 7.58		
Department			<0.001		0.008		0.194	
Surgery	94(30.4)	9.01 ± 2.76		30.35 ± 3.78		36.97 ± 6.88		
Anesthesiology and operating room nurse	163(52.8)	6.66 ± 2.21		31.5 ± 3.58		36.2 ± 7.44		
The relevant internal medicine	52(16.8)	7.73 ± 4.08		29.92 ± 4.26		34.54 ± 9.94		
Work experience (years)			0.456		0.519		0.817	
<10	108(35.0)	9.32 ± 2.91		30.56 ± 4.13		36.22 ± 7.77		
[10, 20]	104(33.7)	9.22 ± 2.88		31.15 ± 3.76		36.45 ± 8.03		
>20	97(31.4)	8.85 ± 2.74		30.96 ± 3.47		35.76 ± 7.55		
Hospital classification			0.050		0.567		0.131	
Public primary hospital	12(3.9)	7.00 ± 4.00		29.17 ± 4.78		31.75 ± 11.62		
Public secondary hospital	69(22.3)	8.91 ± 2.76		30.80 ± 3.38		36.12 ± 7.49		
Public tertiary hospital	216(69.9)	9.37 ± 2.71		30.98 ± 3.89		36.50 ± 7.38		
Specialized hospital	10(3.2)	8.40 ± 4.03		31.60 ± 3.53		36.00 ± 8.11		
Private hospital	2(0.6)	8.50 ± 0.71		31.00 ± 5.66		27.00 ± 24.04		
Participated in the training of PND			0.002		0.006		0.006	
Yes	90(29.1)	9.91 ± 2.37		31.81 ± 4.01		38.04 ± 5.37		
No or unclear	219(70.9)	8.82 ± 2.97		30.51 ± 3.66		35.38 ± 8.45		

Furthermore, the knowledge scores varied for participants with different ages (P = 0.017) and departments (P < 0.001). These findings suggest that both age and departmental context play crucial roles in the knowledge levels among participants. The healthcare professionals over 40 years old have a better knowledge of PND than those aged between 30 and 40. This might be related to their greater clinical experience. It is worth noting that the PND knowledge level of the healthcare professionals in the surgical department is higher than that of the healthcare professionals in the internal medicine department, who have access to surgical patients. This indicates that the medical staff in the surgical department is more concerned about the occurrence of PND. However, the PND knowledge level of anesthesiologists and operating room nurses is lower than that of the medical staff in the internal medicine department, who have access to surgical patients. This suggests that anesthesiologists and operating room nurses urgently need training to enhance their PND knowledge. In terms of attitude scores, there were differences among participants with different ages (P = 0.002), education (P = 0.006), occupation type (P < 0.001), profession title (P = 0.007), and department (P = 0.008). These results indicate that multiple demographic characteristics influence participants’ attitudes, which may have implications for targeted interventions or educational strategies. Among them, the attitude scores of healthcare professionals under the age of 30 towards PND were lower than those of healthcare professionals aged 30–40 and those over 40. This suggests that younger doctors (under the age of 30) need more training on PND, understanding its hazards, as well as the importance of prevention and treatment of PND, and they will have a more positive attitude. The attitudes of medical staff with associated or lower educational degrees towards PND were significantly lower than those of medical staff with master’s or higher degrees. The attitude scores of healthcare workers with junior or lower professional titles towards PND were significantly lower than those of healthcare workers with associate vice or higher professional titles. Furthermore, the attitude scores of surgical medical staff, anesthesiologists, and operating room nurses regarding PND were all higher than those of internal medicine medical staff who have access to surgical patients.

Conversely, the study found no significant differences in practice scores when evaluated by demographic characteristics, indicating a uniformity in practice levels across the analyzed groups. This suggests that factors influencing knowledge and attitude may not directly translate into variations in practice. Furthermore, participants who had undergone training in PND exhibited significantly higher knowledge and practice scores compared to those who did not receive such training. Specifically, the average knowledge score for the trained group was 9.91 (±2.37), while the untrained group’s average was 8.82 (±2.97), resulting in a significant p-value of 0.002. Similarly, the trained participants achieved a practice score of 38.04 (±5.37), in contrast to the untrained group’s score of 35.38 (±8.45), with this comparison yielding a p-value of 0.006. It is noteworthy that the attitude scores of participants who had participated in PND training (31.81 ± 4.01) differ from those who had not (30.51 ± 3.66). These results suggested that training could influence knowledge, attitudes, and practice significantly.

The analysis of knowledge dimensions revealed several critical areas where misconceptions exist among participants regarding perioperative neurocognitive disorders (PND). Specifically, items scoring below 70% on correctness highlighted significant gaps in understanding. For instance, only 52.1% of respondents recognized that neuropsychological testing is currently considered the “gold standard” for diagnosing PND in clinical practice. This indicates a widespread lack of awareness regarding the established diagnostic methods for this condition. Furthermore, 60.2% of participants incorrectly thought that PND is postoperative delirium, illustrating a misunderstanding of the concept and classification associated with PND. Moreover, 63.4% of the individuals surveyed mistakenly believed that the type of surgery is unrelated to the occurrence of perioperative neurocognitive disorders (PND), underscoring a critical need for enhanced education regarding the factors influencing postoperative cognitive health. All other items were correctly answered by ≥70% of the participants (Table S1), reflecting a generally adequate understanding of the predominant concepts related to PND. In terms of attitudes towards PND, the data exhibited a strong consensus among participants, with 97.09% affirming the necessity for healthcare professionals to actively address PND. This overwhelming agreement highlights the recognition of PND as a significant concern within surgical practices, necessitating appropriate attention and intervention. However, it is particularly revealing that 27.51% of participants believed that the responsibility for preventing PND predominantly falls on anesthesiologists, rather than being a shared responsibility among attending physicians and nursing staff (Table S2). This perception may lead to disparities in collaborative approaches to PND prevention and suggests a need for improved communication and teamwork among all healthcare providers involved in perioperative care.

Considering the practice toward PND, 81.56% of participants claimed that they would actively take measures to prevent the occurrence of PND when a patient has high-risk factors for developing PND. This demonstrates a strong awareness and proactive stance among healthcare professionals regarding the significance of early intervention in such high-risk situations. In stark contrast, only a small fraction, precisely 1.94% of participants, expressed that they would never take measures under such circumstances. This disparity highlights a positive practice for the prevention of PND among the majority of respondents. Furthermore, when asked about their willingness to enhance their professional skills related to PND, an encouraging 71.20% of participants expressed enthusiasm for participating in relevant training courses, should such opportunities arise (Table S3). This reflects an overall desire for continued professional development and a recognition of the importance of being well-equipped to comprehensively address PND. The findings underscore a positive practice within the participants to engage in learning that could ultimately benefit patient outcomes related to reducing the occurrence of PND in the perioperative period.

The analysis of Pearson’s correlation indicated a positive relationship between the knowledge scores and both the attitude (r = 0.464, P < 0.01) and the practice (r = 0.312, P < 0.01) scores. This finding suggests that higher levels of knowledge are associated with more favorable attitudes and enhanced practices. Additionally, a positive correlation was found between the attitude scores and the practice scores (r = 0.240, P < 0.01) (Table 2), which indicates that improvements in attitudes are likely to lead to improvements in practices. The path analysis revealed that the model exhibits a highly favorable fit, as indicated by the fit indices. This suggests that the model is well-fitting (Table 3). The SEM results showed that participants’ knowledge of PND for healthcare workers directly and positively affected their attitudes (β = 0.545, P < 0.001) and practices towards PND (β = 0.230, P = 0.002), while the attitudes towards PND also directly and positively affected their practices (β = 0.238, P = 0.001). In addition, the knowledge exerted an indirect effect on practice through attitudes (β = 0.581, P = 0.003) (Table 4 and Fig. 1).

Table 2 Pearson’s correlation analysis.

	Knowledge	Attitude	Practice	
Knowledge	1			
Attitude	0.464 (P < 0.01)	1		
Practice	0.312 (P < 0.01)	0.240 (P < 0.01)	1	

Table 3 Model fit.

Indicators	Reference	Results	
CMIN/DF	1–3 Good	2.709	
RMSEA	<0.08 Good	0.074	
RMR	<0.08 Good	0.081	
CFI	>0.80 Good	0.854	
TLI	>0.80 Good	0.841	
GFI	>0.80 Good	0.820	
AGFI	>0.80 Good	0.789	
PGFI	>0.50 Good	0.701	

Table 4 The results of structural equation modeling (SEM).

			Estimate(β)	P	
Attitude	<–	Knowledge	0.982	<0.001	
Practice	<–	Knowledge	1.031	0.002	
Practice	<–	Attitude	0.592	0.001	

Figure 1 The structural equation model of health workers’ KAP regarding PND.

Discussion

Perioperative neurocognitive disorders have adverse effects on the recovery and quality of life improvement, as well as long-term survival of patients scheduled for elective surgeries, especially old populations patients (Evered et al., 2018). It has now become a public health issue and is receiving worldwide attention (Khachaturian et al., 2020). Healthcare workers play a crucial role in the prevention and treatment of PND. Investigating the knowledge, attitudes, and practices of healthcare workers regarding PND has practical clinical significance for preventing the occurrence of PND and treating it.

Medical personnel possess a comprehensive understanding of PND and generally demonstrate favorable attitudes toward managing this condition. They actively engage in practices aimed at mitigating its occurrence. Despite these strengths, specific areas require further enhancement. The findings of this study are crucial for informing the design of specialized training programs tailored for medical staff. These programs aim to equip healthcare providers with the necessary skills to effectively implement appropriate medical and nursing interventions during the perioperative phase. By doing so, the goal is not only to decrease the occurrence of PND but also to improve the overall efficacy of its treatment.

Healthcare professionals who have previously participated in PND training may exhibit superior knowledge, attitudes, and practice scores compared to their counterparts who have not undergone such training. Consistent with our findings, previous studies have suggested that knowledge of PND increases with experience (Hu et al., 2024). The results indicate that participation in PND training plays a significant role in enhancing the knowledge and clinical practice of healthcare professionals. However, the lack of continuous training and insufficient depth of the training may be potential reasons for the unchanged attitudes of trained healthcare professionals. Previous research has demonstrated that continuing education is essential for enabling professionals to keep pace with the ever-evolving medical knowledge (Vieira et al., 2021). Therefore, interventions should ensure the provision of continuous and in-depth training to cultivate a proactive attitude among healthcare professionals.

In terms of the knowledge dimension, the results indicated that 11% of healthcare professionals were still unclear about the classification of PND, and 47.9% were unaware of the diagnostic criteria for PND. This indicates that healthcare professionals still need to acquire a clear understanding of the basic knowledge of PND. Notably, anesthesiologists and operating room nurses exhibited a particularly pronounced deficiency in relevant knowledge about PND, which aligns with previous research findings (Luo et al., 2024). This suggests that anesthesiologists and operating room nurses have a greater need for PND training. Research has demonstrated that only accurate knowledge can lead to appropriate actions (Manns, 2021), and anesthesiologists play a critical role in the prevention of PND (Touchard et al., 2020). Therefore, hospitals must implement targeted interventions aimed at enhancing the knowledge of anesthesiologists and operating room nurses to improve the prevention, identification, and treatment of PND.

Most healthcare workers maintain a positive attitude towards PND; however, some participants believe that the prevention of PND relies solely on anesthesia personnel and that enhancing PND management cannot prevent its occurrence. These attitudes indicate that certain healthcare professionals are not sufficiently proactive. Specifically, participants with young medical staff under the age of 30, lower education levels, nurses, and those with lower professional titles exhibit less positive attitudes towards PND, which aligns with previous research findings indicating that PND attitudes increase with experience and professional rank (Hu et al., 2024). Furthermore, surgical medical staff, anesthesiologists, and operating room nurses demonstrate a more positive attitude towards PND, suggesting that as healthcare professionals achieve higher education and professional titles, and gain more exposure to PND patients in their work, they may develop a better understanding of the potential harm PND can inflict on patients and receive more training and evaluation related to PND. Consequently, they exhibit a more positive attitude towards PND. In summary, these results indicate the necessity for targeted training among different populations to foster a proactive and positive mindset among healthcare professionals (Noroozi et al., 2022), thereby collaboratively preventing the occurrence of PND.

In our study, we found no significant differences between most healthcare worker characteristics and their PND practice scores. However, it is encouraging that healthcare workers who participated in PND training demonstrated improved practices, underscoring the importance of such training. While most healthcare workers assess patients’ cognitive status and take preventive measures for high-risk PND patients, fewer adopt treatment practices, such as the use of non-opioid analgesics to manage pain. This indicates that the training content should place greater emphasis on treatment measures for PND and strengthen the focus on factors influencing the severity of PND.

Correlation and SEM analyses indicate a significant positive correlation among healthcare professionals’ KAP. Higher levels of knowledge are associated with positive attitudes and effective practices towards PND. Furthermore, knowledge directly influences attitudes, while both knowledge and attitudes directly impact practices. Knowledge can also indirectly influence practice by influencing attitudes. This finding aligns with existing research on KAP in the medical field (Afaya et al., 2022; Cai et al., 2024). Recognizing these relationships is essential for enhancing training methods. Tailored training approaches should be developed for different populations to improve healthcare professionals’ understanding of PND, ultimately enhancing their attitudes and practices.

This study also has several limitations. One limitation of this study is that the questionnaire did not strictly distinguish between the practices of elective surgery and emergency surgery. In the future, our research will focus on the differences in PND between elective surgeries and emergency surgeries. Moreover, this study only focused on the KAP scores of doctors (including anesthesiologists) and nurses, but ignored other health professionals. In the future, our research will expand to include more types of healthcare providers. And we lack a questionnaire development team, thus unable to calculate the surface validity and content validity. Although we conducted a multi-center study, it was primarily confined to Shandong Province in China. This convenience sampling approach limits the generalizability of our findings. The socio-economic, cultural, and ethnic characteristics of our cohort may not be representative of the broader Chinese population suffering from this condition. Consequently, the associations we observed might be stronger or weaker in other settings, and the extrapolation of our conclusions should be made with caution. Future multi-center studies spanning diverse geographic and cultural contexts are needed to validate our results.

Finally, although the final sample size of 309 met and exceeded minimum requirements for the statistical analyses conducted, it should be noted that it was slightly below the initial target of 336. This was primarily due to a lower-than-anticipated survey response rate within the study timeframe. While this shortfall is not believed to substantially impact the validity of our findings, future studies may aim for larger samples to enhance generalizability. Furthermore, while we distributed questionnaires across hospitals at all levels, the proportion of participants from tertiary hospitals was relatively high, potentially introducing bias into the results. Additionally, this study relies on self-reported data, which may lead to response bias, including recall bias, social desirability bias, and subjective interpretation. Participants may have inaccurately recalled past behaviors or may have provided answers they perceived as socially acceptable (Latkin et al., 2017). This error may lead to overstatement in their KAP scores, which could potentially make the observed differences smaller and even result in false correlations in some cases. Although objective measurement methods would be more ideal, they are not feasible within the scope of this study. Therefore, our research results should be interpreted in conjunction with this potential measurement error.

Conclusions

In summary, healthcare professionals may possess sufficient knowledge and have positive attitudes and proactive practices regarding PND. Training significantly impacts their knowledge, attitudes, and practices. Therefore, targeted training is necessary to enhance healthcare professionals’ knowledge, attitudes, and practices concerning PND, which may improve the prevention and treatment of PND.

Supplemental Information

Supplemental Information 1 Correctness of knowledge

Supplemental Information 2 Attitudes of healthcare workers regarding PND, n (%)

Supplemental Information 3 Distribution of practice (%)

Supplemental Information 4 The results of multiple comparisons of knowledge scores

Supplemental Information 5 The results of multiple comparisons of attitude scores

Supplemental Information 6 Cronbach’s Alpha

Supplemental Information 7 Questionnaire

Supplemental Information 8 Raw data

Supplemental Information 9 Codebook

Supplemental Information 10 Translations for Chinese in raw data

Supplemental Information 11 STROBE statement

Additional Information and Declarations

Competing Interests

Author Contributions

Human Ethics

Data Availability

The authors declare there are no competing interests.

Ke Zhang conceived and designed the experiments, performed the experiments, analyzed the data, prepared figures and/or tables, authored or reviewed drafts of the article, and approved the final draft.

Xuefeng Gu conceived and designed the experiments, performed the experiments, analyzed the data, prepared figures and/or tables, and approved the final draft.

Xiaoru Li conceived and designed the experiments, performed the experiments, analyzed the data, prepared figures and/or tables, and approved the final draft.

Honghai Peng conceived and designed the experiments, performed the experiments, analyzed the data, prepared figures and/or tables, authored or reviewed drafts of the article, and approved the final draft.

The following information was supplied relating to ethical approvals (i.e., approving body and any reference numbers):

The Ethics Committee review of Central Hospital Affiliated to Shandong First Medical University approved the study (202501200008).

The following information was supplied regarding data availability:

The raw data is available in the Supplementary File.

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
