# Peer review of "Knowledge, attitude, and practice toward perioperative neurocognitive disorders among healthcare workers in Shandong, China: a cross-sectional study"

_PeerJ, doi:10.7717/peerj.20450_

## Round 0.1 · original submission · Major Revisions

· Academic Editor

Major Revisions

·

Basic reporting

I appreciate the opportunity to read this interesting article. It is an interesting articles that highlights what healthcare workers know about perioperative neurocognitive disorder. This study is survey study with excellent response rate. The findings are straightforward more or less. I do have a few suggestions to improve clarity and reproducibility of the article.
• Introduction – It would be more helpful to describe what this neurocognitive disorder looks like. Perhaps a case example or two. If the authors don’t want to put it in a narrative, perhaps in a table format of what it looks like in an older person and younger person with antecedents, presentation of symptoms, treatment, resolution. I imagine this can manifest in many ways. For those of us not in the hospital setting but still in a clinical setting, it would be helpful to know what this looks like realistically.
• Good job – Line 74 – I was about to mention what constitutes a healthcare worker but these authors anticipated my question and provided an excellent
• Items for the questionnaire/survey are not clear to me. I need to see the actual items. Could this be included in an appendix? In fact, it might be helpful to know how people responded to each item.
• I love the SEM. Why was not GFI, AFGI, PGFI included as fit statistics?

Experimental design

This is a basic survey design. It has an excellent response rate. There are no ethical concerns. But it would be VERY helpful to see the items of the survey; in the current version of the article, the items are loosely described which is not descriptive enough for me.

Validity of the findings

It is hard for me to judge the validity of the findings without being able to see the individual survey items that made up the scores for knowledge, attitude, and practice.

Additional comments

I think the article/study does not over state the findings, but more details are needed.

Reviewer 2 ·

Basic reporting

See attached

Experimental design

See attached

Validity of the findings

See attached

Additional comments

See attached

Annotated reviews are not available for download in order to protect the identity of reviewers who chose to remain anonymous.

·

Basic reporting

Introduction
1. The introduction is informative but contains overly dense sentences; for example, the long list of interventions (lines 48–51) should be broken into shorter sentences for better clarity and readability.
2. While the introduction defines perioperative neurocognitive disorders, it does not clearly explain the differences between its subtypes—such as postoperative delirium and postoperative neurocognitive disorder. These conditions differ in timing, symptoms, and diagnostic criteria. Adding a brief explanation of each subtype would enhance clarity and help readers understand the scope and complexity of PND.
3. The authors note prior focus on anesthesiologists but offer limited justification for including other healthcare workers (e.g., nurses, surgeons, operating room technicians). Without clarifying these groups’ roles in PND prevention, the rationale for their inclusion feels weak. This gap undermines the study’s scope and claims novelty. A clearer explanation of their relevance is needed.
4. The rationale for selecting Shandong Province is unclear; adding regional context would strengthen the study’s relevance and justification.

Experimental design

Materials & Methods
1. The manuscript does not mention adherence to the STROBE guidelines, nor does it include a completed STROBE checklist. Including this would ensure transparent and standardized reporting, especially for an observational, cross-sectional design.
2. The study lacks a detailed description of participant recruitment and flow (STROBE item 13), such as how many individuals were approached, how many declined, and how many were excluded after applying the data quality criteria. A flow diagram would greatly improve clarity and reproducibility.
3. STROBE recommends discussing sources of potential bias (item 9), but the manuscript does not address important biases inherent in convenience sampling or self-reported data, such as selection bias, social desirability bias, and non-response bias.
4. While the pilot study reports satisfactory internal consistency (Cronbach’s alpha), the manuscript lacks information on content and construct validity assessments or expert review of the questionnaire. Internal consistency alone is insufficient to establish the validity and robustness of the measurement instrument.
5. While the exclusion of rapid responses and logically inconsistent answers is commendable, the threshold of 90 seconds is arbitrary and lacks justification. A more evidence-based or pretested time threshold would strengthen the rigor of the data cleaning process.
6. The explanation of study size is limited to a minimum sample size calculation based on the number of predictors, but it lacks a detailed justification considering expected response rates, effect sizes, or power analysis. A more comprehensive sample size determination would strengthen the study’s methodological rigor and credibility.

Validity of the findings

Discussion
1. The limitations are mentioned superficially without discussing how sample bias, regional confinement, and self-reporting might significantly skew results. A more in-depth critical appraisal is necessary to contextualize the findings and temper conclusions regarding generalizability and reliability.

---

## Round 0.2 · Minor Revisions

· Academic Editor

Minor Revisions

Thank you for submitting the revised version of your manuscript. We appreciate the careful attention you have given to the reviewers’ comments. The paper is much improved and close to being suitable for publication.

However, before acceptance, a few issues must be addressed.

1. There seems to be inconsistency in the number of items (Attitude dimension). In the abstract, you report 10 items for the attitude scale, whereas in the main text you describe 7 items (with a score range of 7–35). Please reconcile this discrepancy and ensure that the description of the questionnaire is consistent across all sections.

2. There also seems to be a discrepancy in reported practice scores. In the abstract, the mean practice score is reported as 36.16 ± 7.77, but in the Results section it appears as 26.16 ± 7.77. Given the scale range (9–45), a score of 36 is more consistent with your interpretation of positive practices, while 26 suggests a more neutral level. Please clarify which is correct and ensure consistency across the manuscript.

3. In terms of the the sample size justification, the methods state that a target sample size of 336 was calculated (accounting for 20% non-response). However, the final valid sample was 309. While this still exceeds the minimum required based on the “10 per variable” rule, the shortfall relative to the original plan should be explicitly acknowledged in the limitations.

4. In the reporting of the SEM results, the reported path coefficients (e.g., β = 0.982 and β = 0.912) are unusually high. While this may be correct, the SEM results need to be supported with fit indices (e.g., CFI, TLI, RMSEA, χ²/df). Please provide these indices to demonstrate that the model is well fitting.

5. There is inconsistency in the terminology. Please standardise the use of “perioperative neurocognitive disorder (PND)” versus “postoperative neurocognitive disorder (POCD).” At present, both are used interchangeably, which is confusing for readers.

There are also several minor editorial issues
A) Check references and in-text citations for spacing and formatting consistency (e.g., “mortality(Yang et al. 2022)” should read “mortality (Yang et al. 2022)”).
B) Some sentences would benefit from light language polishing for clarity.

Please ensure that these issues are corrected, then the manuscript will be ready for publication.

Reviewer 3 ·

Basic reporting

The manuscript now adheres well to the principles of transparent and complete reporting. The authors have incorporated a STROBE checklist and made targeted revisions to improve clarity and readability. Key improvements include:

1. Rewriting overly dense sections of the introduction to enhance readability.

2. Clearly defining the subtypes of perioperative neurocognitive disorders (PND) for improved conceptual understanding.

3. Justifying the inclusion of multidisciplinary healthcare providers in the study scope, thereby enhancing the relevance and novelty of the work.

Additionally, the discussion has been expanded to include a critical appraisal of limitations such as self-reporting bias and regional confinement. These changes improve both the transparency and contextualization of the study findings.

Experimental design

The study design—a cross-sectional, questionnaire-based survey—is appropriate for the stated objectives. The authors have responded to prior methodological concerns by:

1. Clarifying the recruitment process and participant flow, including the total number approached, excluded, and analyzed.

2. Justifying the data exclusion threshold using pretest data.

3. Acknowledging the lack of formal validity testing (e.g., content or construct validity) for the survey instrument, while explaining resource limitations.

4. Providing a more comprehensive rationale for sample size considerations, including an estimated non-response rate.

Validity of the findings

The findings are presented appropriately and interpreted with caution. The authors have significantly strengthened the discussion of potential biases, including:

1. Selection bias due to convenience sampling.

2. Social desirability and recall biases inherent in self-reported data.

3. Regional limitations affecting generalizability to broader populations.

By framing their conclusions within these constraints and calling for future studies in more diverse regions and with objective data collection methods, the authors demonstrate scientific responsibility and integrity.

Despite some limitations in instrument validation, the study’s conclusions about the knowledge, attitudes, and practices (KAP) of multidisciplinary teams in PND prevention are supported by the data and offer novel insights for clinical practice and policy.

---

## Round 0.3 · accepted · Accept

· Academic Editor

Accept

The authors have now fixed all the minor issues noted, so the paper is ready for publication.